# Voltage controlled interfacial magnetism through platinum orbits

Shinji Miwa[1,2,*], Motohiro Suzuki[3,*], Masahito Tsujikawa[4,5,*], Kensho Matsuda[1,*], Takayuki Nozaki[6], Kazuhito Tanaka[1], Takuya Tsukahara[1], Kohei Nawaoka[1], Minori Goto[1,2], Yoshinori Kotani[3], Tadakatsu Ohkubo[7], Frédéric Bonell[1], Eiiti Tamura[1], Kazuhiro Hono[7], Tetsuya Nakamura[3], Masafumi Shirai[4,5], Shinji Yuasa[6] & Yoshishige Suzuki[1,2,6,7]

Electric fields at interfaces exhibit useful phenomena, such as switching functions in transistors, through electron accumulations and/or electric dipole inductions. We find one potentially unique situation in a metal–dielectric interface in which the electric field is atomically inhomogeneous because of the strong electrostatic screening effect in metals. Such electric fields enable us to access electric quadrupoles of the electron shell. Here we show, by synchrotron X-ray absorption spectroscopy, electric field induction of magnetic dipole moments in a platinum monatomic layer placed on ferromagnetic iron. Our theoretical analysis indicates that electric quadrupole induction produces magnetic dipole moments and provides a large magnetic anisotropy change. In contrast with the inability of current designs to offer ultrahigh-density memory devices using electric-field-induced spin control, our findings enable a material design showing more than ten times larger anisotropy energy change for such a use and highlight a path in electric-field control of condensed matter.

[1] Graduate School of Engineering Science, Osaka University, 1-3 Machikaneyama, Toyonaka 560-8531, Japan. [2] Center for Spintronics Research Network (CSRN), Osaka University, Toyonaka 560-8531, Japan. [3] Japan Synchrotron Radiation Research Institute (JASRI), Sayo 679-5198, Japan. [4] Research Institute of Electrical Communication, Tohoku University, Sendai 980-8577, Japan. [5] Center for Spintronics Research Network (CSRN), Tohoku University, Sendai 980-8577, Japan. [6] National Institute of Advanced Industrial Science and Technology (AIST), Spintronics Research Center, Tsukuba 305-8568, Japan. [7] Research Center for Magnetic and Spintronic Materials, National Institute for Materials Science (NIMS), Tsukuba 305-0047, Japan. * These authors contributed equally to this work. Correspondence and requests for materials should be addressed to S.M. (email: miwa@mp.es.osaka-u.ac.jp).

Spin transfer torque has surpassed current-induced magnetic fields as the preferred technology for switching magnetization directions in nanoscale magnets because of its low energy consumption and excellent scalability[1]. It has been used as the operational technology for nonvolatile random access memory[2] and microwave devices[3–5] based on magnetic tunnel junctions[6]. However, spin transfer switching exhibits Joule heating that remains too large to ignore. The energy required for magnetization switching is more than $10^7$ times the Landauer limit of $k_B T \ln 2$, where $k_B$ is the Boltzmann constant[7]. Thus, a novel method offering magnetization control without electric current is highly desirable.

An electric field at a surface is known to exhibit useful phenomena, such as the pinch-off phenomenon in field-effect transistors[8], inductions of Mott transitions[9], superconducting phases[10], ferromagnetic phases[11,12] and magnetic anisotropy[13]. All of these phenomena are attributable to one or both of two factors: electron accumulations or electric dipole inductions[14]. In particular, electric-field control of magnetic properties at room temperature attracts much attention because of its great potential for enabling the construction of ultralow-power-consumption electric devices[15–43]. Voltage-controlled magnetic anisotropy (VCMA) in Fe-MgO-based magnetic tunnel junctions[20] has shown that magnetization of nanomagnets can be controlled in fast periods (down to 0.1 ns) by electric fields, as indicated by a bistable precessional magnetization switching[23,27,40] and ferromagnetic resonance excitation[25,26,31]. One mechanism to explain magnetic anisotropy of magnetic atoms at surfaces is the magnetization direction dependence of its orbital angular momentum because of spin–orbit interactions[44]. Therefore, one explanation of VCMA is selective electron–hole doping into a particular electron orbital of magnetic metal atoms at the interface. Because the above-described mechanism is purely electronic, VCMA can be an ultimate technology for the operation of spintronics devices, such as nonvolatile random access memory[38], where high-speed operation with high writing endurance is indispensable. So far, such electronic VCMA is reported[31,35,36] to be $< 0.1 \, pJ \, V^{-1} \, m^{-1}$. This value is too small to switch the magnetic nanopillar possessing adequate thermal stability for the memory. In contrast, reports have shown that charge doping by an external electric field induces reversible electrochemical reactions and quite large values of VCMA $(> 10 \, pJ \, V^{-1} \, m^{-1})$[33,34]. This electrochemical VCMA is reported in certain systems, such as $FeCoO_x|MgO$ (ref. 28), FePt|electrolyte[29], $Fe|BaTiO_3$ (ref. 32) and $Co|GdO_x$ (refs 33,34). Such large values of electrochemical VCMA seem attractive but require thermal activation processes[34]. Because electrochemical VCMA presents hysteretic effects with limited operation speed less than the sub-millisecond range, it lies beyond the scope of this study.

Here we show a mechanism to enhance electronic VCMA. We focus on a unique property of an electric field at a metal–dielectric interface, at which the electric field is inhomogeneous along the perpendicular direction at the atomic scale. This inhomogeneity arises from the strong electrostatic screening effect in metals. Such an inhomogeneous electric field couples not only with an electric dipole but also a quadrupole of an electron shell in a metal layer. The electric quadrupole coupled with the intra-atomic magnetic dipole should directly influence magnetic anisotropy[45]. In this paper, we prepare monatomic layers of Pt at an Fe–MgO interface by molecular-beam epitaxy in an ultrahigh vacuum. Since the Pt atoms have a considerable amount of spin polarization because of their hybridization with Fe and large spin–orbit interactions, the influence of the modified electron orbitals of Pt on the level of magnetocrystalline anisotropy energy (MAE) should be significant. *In situ* X-ray magnetic circular dichroism (XMCD) spectroscopy reveals that an external voltage induces a finite magnetic dipole moment of Pt. Our theoretical analysis shows that the monatomic Pt layer at the Fe–MgO interface makes the dominant contributions to the MAE in the system and its voltage-induced change. We also find that a voltage induction of the magnetic dipole moment, originating from spin-flip excitation between the exchange-split majority and minority spin bands, dominates the voltage-induced MAE change. This mechanism differs from the aforementioned orbital anisotropy induced by charge doping. Our finding enables the design of novel materials showing electronic VCMA larger by more than a factor of 10 for practical applications.

## Results

**Experimental design**. An $L1_0$-FePt|MgO system has been prepared to conduct XMCD spectroscopy. First, a controlled experiment was conducted to optimize the structure of $L1_0$-FePt|MgO (see Method 1, Supplementary Note 1 and Supplementary Figs 1 and 2). As suggested by the previously reported theoretical study[18], the VCMA in the FePt|MgO with Pt–MgO interface $(0.14 \, pJ \, V^{-1} \, m^{-1})$ was several times larger than that with the Fe–MgO interface $(0.03 \, pJ \, V^{-1} \, m^{-1})$. Moreover, more than one monatomic layer of Fe is necessary to induce a ferromagnetic property in the FePt. Hence, the FePt ferromagnetic layer consisting of two monatomic layers of Fe and Pt, which has a Pt–MgO interface, was designed as shown in Fig. 1a,b. Figure 1c shows a high-angle annular dark-field scanning transmission electron microscopy image of the sample (see Method 2, Supplementary Note 2 and Supplementary Fig. 3). This image indicates that the FePt is epitaxially grown on $Pd(0\,0\,1)$ and a monoatomic Pt layer lies between the Fe and MgO. We studied voltage-induced changes in the electronic and magnetic states of Pt, which exhibits a large spin–orbit interaction and should be responsible for the large VCMA in the system. Using the experimental setup shown in Fig. 1b, we applied external voltages of $\pm 200 \, V$ to the FePt|MgO tunnel junction. The external voltage is equivalent to an external electric field of $\pm 0.7 \, V \, nm^{-1}$ in the MgO dielectric. Such an electric field magnitude is often used in MgO-based magnetic tunnel junctions[23]. X-ray absorption spectroscopy (XAS) and its XMCD spectra were recorded *in situ* at the Pt $L_3$ and $L_2$ edges by detecting the X-ray fluorescence yields (see Method 3). In this configuration, positive external voltages induce electron accumulations at the Pt–MgO interface.

**Synchrotron XAS**. Figure 1d shows the element-specific magnetization hysteresis curve of the Pt. The depicted data were recorded by monitoring the XMCD signals at the $L_3$ edge of the Pt (11.569 keV) as a function of the magnetic field applied perpendicularly to the film. The loops for negative (blue circles; $-200 \, V$) and positive (red triangles; $+200 \, V$) external voltages demonstrate voltage-induced significant modifications to the magnetic properties of the Pt. The existence of the hysteresis shows that the FePt|MgO film is perpendicularly magnetized. At the Pt $L_3$ edge, the negative XMCD signals under positive magnetic fields show that the magnetic moment of the Pt was parallel to the magnetic field direction, that is, the magnetic moment of the Pt exhibited ferromagnetic coupling with the magnetic moment of the Fe. The coercive field was large at negative voltages, corresponding to the enhanced perpendicular magnetic anisotropy. Therefore, electron depletion at the Pt–MgO interface enhanced the perpendicular magnetic anisotropy. This VCMA polarity matched the result of experiments with Fe-MgO-based magnetic tunnel junctions[17,20,23–27,36].

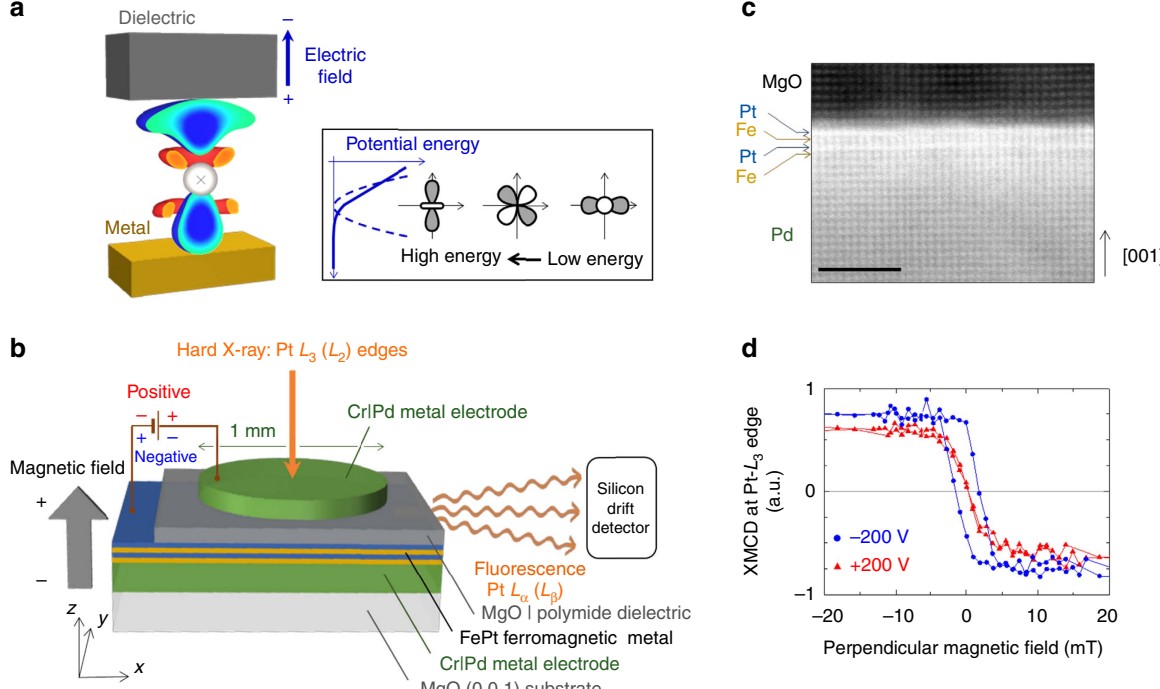

**Figure 1 | Experimental design and VCMA at Pt–MgO interface.** (**a**) A sectional view of a Pt atom with its voltage-induced charge distribution from the first-principles study. Blue and red depict the hole accumulation and depletion, respectively. The charge accumulation and electric quadrupole induction are distinct when compared to the electric dipole induction. This characteristic is a unique property of the electric field at the metal–dielectric interface. The electric quadrupole is coupled with the magnetic dipole and changes the magnetic anisotropy through spin–orbit interactions. (**b**) A schematic depiction of the device structure and the experimental setup of *in situ* synchrotron XAS. The device consists of Cr|Pd metal electrodes, MgO|polyimide dielectric material and FePt ferromagnetic metal layer. The FePt consists of two monatomic layers of Fe and Pt. (**c**) A high-angle annular dark-field scanning transmission electron microscopy image. Scale bar, 2 nm. (**d**) Element-specific magnetization hysteresis curves from the XMCD signal at the $L_3$ edge energy of Pt under external voltages of $\pm 200$ V. The external voltage is equivalent to an external electric field of $\pm 0.7$ V nm$^{-1}$ in the MgO dielectric. Such an electric field magnitude is often used in MgO-based magnetic tunnel junctions[23].

Results of the polarization-averaged XAS and its XMCD are shown in Fig. 2. The measurements were conducted at room temperature. A perpendicular magnetic field of $\pm 60$ mT was applied to saturate the magnetization of the FePt. Figure 2a shows the polarization-averaged XAS around the $L_3$ and $L_2$ energy edges of the Pt. The intensity of the XAS white line observed at the absorption threshold corresponds to the unoccupied density of states of Pt near the Fermi level. Figure 2b illustrates the XAS difference defined by the XAS taken at $+200$ V subtracted from the XAS taken at $-200$ V. Note the distinct change in the XAS, indicating the change in the density of states of Pt because of the external voltage. The black arrow denotes the transition from the $2p$ core to the $5d$ valence of Pt. According to the selection rule, the electric dipole transition at the $L_3$ edge takes place for transitions of $2p_{3/2} \rightarrow 5d_{3/2}$ and $2p_{3/2} \rightarrow 5d_{5/2}$, whereas the $L_2$ edge only involves the $2p_{1/2} \rightarrow 5d_{3/2}$ transition. Because the XAS difference dominates only at the $L_3$ edge, the results indicate that a negative electric field induced the electron depletion at the $5d_{5/2}$ level, which was located at an energy level shallower than $5d_{3/2}$. The red arrow denotes the $2p$ to $6s$ transition[46], which indicates electron doping at $6s$. Because of the electrostatic screening effect, an electron depletes a Pt atom, which is consistent with the phenomenon of electron depletion in the $5d$ orbital. In addition, the electric dipole in a Pt atom can contribute to the screening effect. Because $s$–$p$ hybridization is indispensable for creating electric dipoles, $6s$ electrons increase because of the charge transfer from the $p$ to $s$ orbitals. Figure 2c demonstrates the voltage-induced changes in the XMCD spectra, and Fig. 2d plots the numerical integrals of the XMCD spectra, related to atomic magnetic moments of Pt, over the X-ray energy.

Note that both XMCD intensities at the $L_3$ and $L_2$ edges are enhanced by the application of a negative voltage.

Then, conventional sum-rule analysis can characterize the magnetic moments[47–49]:

$$m_{eff} = m_S - 7m_T = -n_{5d}\mu_B\left(\frac{a_{L3} - 2a_{L2}}{A_{L3} + A_{L3}}\right), \quad (1)$$

$$m_L = -\frac{2}{3}n_{5d}\mu_B\left(\frac{a_{L3} + a_{L2}}{A_{L3} + A_{L2}}\right) \quad (2)$$

and

$$n_{5d} = \frac{1}{C}(A_{L3} + A_{L2}), \quad (3)$$

where $m_{eff} = m_S - 7m_T$, $m_L$ and $n_{5d}$ are the effective spin magnetic moment, orbital magnetic moment and hole number of Pt in its $5d$ orbitals, respectively. Further, $m_S$, $m_T$, $\mu_B$, $a_{L3}$ ($a_{L2}$), $A_{L3}$ ($A_{L2}$) and $C$ with a value of 13.29 eV are the spin magnetic moment, magnetic dipole moment, Bohr magneton, XMCD integral at the $L_3$ ($L_2$) edge, white-line intensity of XAS at the $L_3$ ($L_2$) edge and the proportionality coefficient, respectively. Each moment has the following relations: $m_S = -2\mu_B\langle S_\alpha\rangle/\hbar$, $m_L = -\mu_B\langle L_\alpha\rangle/\hbar$ and $m_T = \mu_B\langle T_\alpha\rangle/\hbar$, where $\langle S_\alpha\rangle$, $\langle L_\alpha\rangle$ and $\langle T_\alpha\rangle$ express the expectation value of the spin angular momentum (**S**), orbital angular momentum (**L**) and magnetic dipole operator (**T**) in the X-ray direction, respectively. When the influence of spin–orbit coupling on $\langle T_\alpha\rangle$ is neglected, the magnetic dipole operator can be described as $\mathbf{T} = \mathbf{S} - 3\hat{\mathbf{r}}(\hat{\mathbf{r}} \cdot \mathbf{S}) \approx -2\mathbf{Q} \cdot \mathbf{S}/7$, where $\hat{\mathbf{r}}$ and **Q** are the position unit vector and electric quadrupole tensor, respectively. For a system with $C_{4v}$

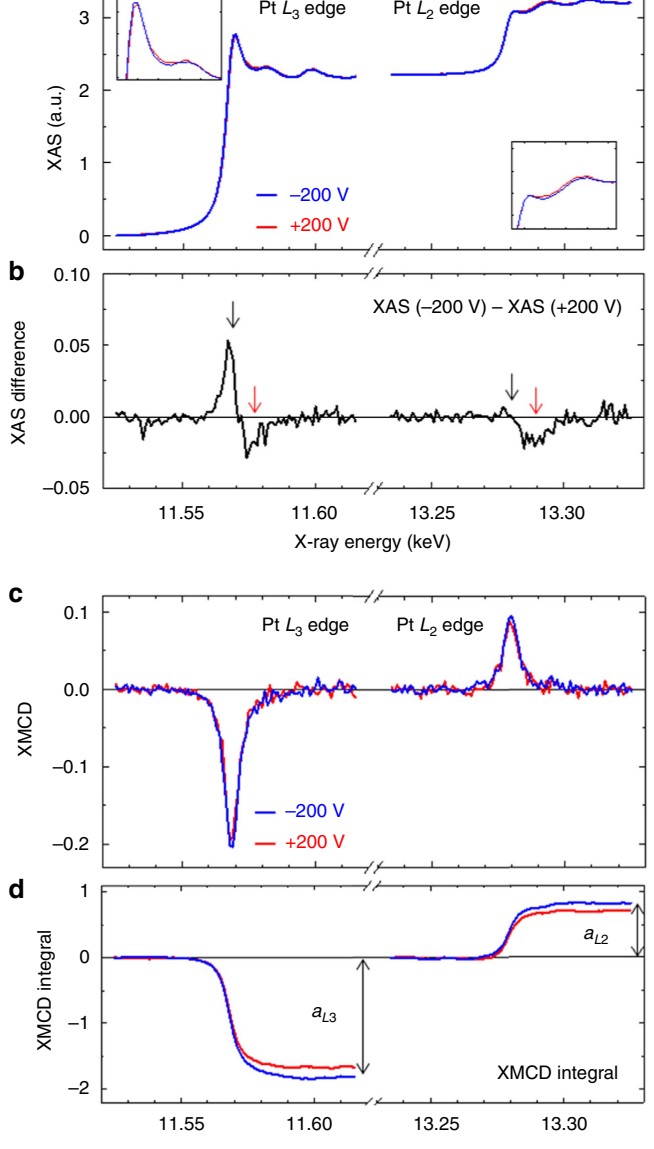

**Figure 2 | Synchrotron XAS.** (**a**) Results of XAS measured at the Pt $L_{2,3}$ edges. The blue and red curves trace the polarization-averaged XAS spectra taken under conditions of $-200$ V and $+200$ V, respectively. The insets show magnified views. (**b**) The XAS spectra at $+200$ V was subtracted from the XAS spectra at $-200$ V. (**c**) XMCD spectra measured at the Pt $L_{2,3}$ edges. (**d**) Traces depicting the numerical integrals of the XMCD spectra. All measurements were taken at room temperature under perpendicularly applied magnetic fields of $\pm 60$ mT, where the magnetization of the FePt was saturated normal to the film plane.

symmetry, as is the case with our system, the above-described equation for **T** can be simplified as

$$\mathbf{T} \approx \frac{Q_{zz}}{7} \begin{pmatrix} S_x \\ S_y \\ -2S_z \end{pmatrix}. \qquad (4)$$

Here the $z$-direction is perpendicular to the film. We experimentally determined the hole number of the FePt compared with that of the bulk Pt ($n_{5d} = 1.80$)[50]. Supplementary Note 1 and Supplementary Fig. 4 provide a definition of the white-line intensity of XAS.

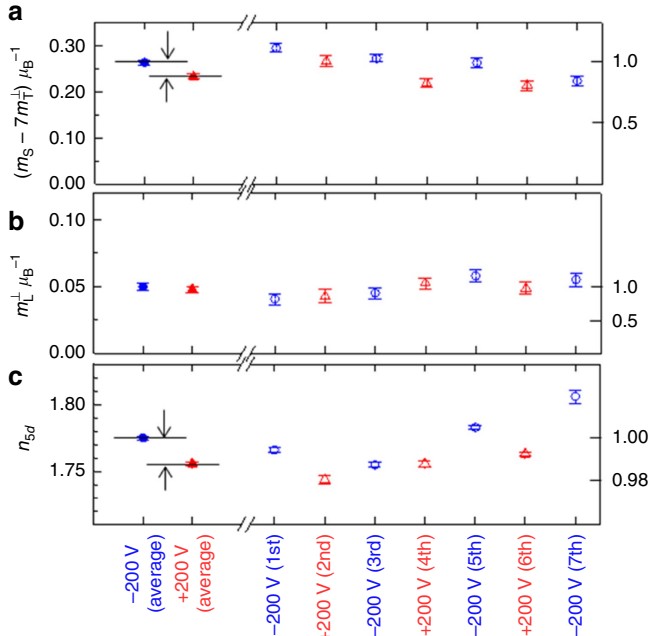

**Figure 3 | Voltage dependence of the magnetic moments and hole numbers of Pt.** (**a**–**c**) The change in the effective spin magnetic moment ($m_S - 7m_T^{\perp}$), the orbital magnetic moment ($m_L^{\perp}$) and the hole number in $5d$ orbitals ($n_{5d}$). Here, $m_S$ and $m_T$ denote the spin magnetic moment and magnetic dipole moment, respectively. The error bars show precision in measurements defined as the s.d. of the numerical integrals of the X-ray magnetic circular dichroism spectra in Fig. 2d.

**Voltage dependence of the magnetic moments.** Figure 3 shows the voltage dependence of the magnetic moments and the hole number of the Pt. We conducted XAS/XMCD measurements seven times. The closed blue circles and closed red triangles show the average results determined using the average spectra appearing in Fig. 2. The open blue circles and open red triangle show each measurement result. The values taken from each measurement gradually varied as time passed. With such variations, distinct zigzag changes corresponding to positive and negative external voltages can be superimposed. As accuracy, errors from true physical parameters under the sum-rule assumption may be $\sim 10\%$. However, errors from relative changes in measurements, precision, are much smaller than the accuracy errors and are displayed as the error bars in Fig. 3 and Table 1. From each result, the zigzag changes are clearly seen for the effective spin magnetic moment and hole number, whereas the orbital magnetic moment lies within the range of error. The averaged results confirm clear external voltage inductions of the effective spin magnetic moment and hole number. Even in the averaged results, voltage applications do not contribute significant differences to the orbital magnetic moment. The effective spin magnetic moment increased by 13%, and the hole number of Pt in the $5d$ orbital increased by 0.019 when the external voltage was switched from positive to negative. Note that the change in the effective spin magnetic moment (13%) was an order of magnitude larger than the change in the hole number (1%). Therefore, the change cannot be explained by the voltage-induced electrochemical reaction resulting in substantial changes in a valence state. In addition, the change should be attributed to the electron redistribution in the Pt atom.

The magnetic moments and their voltage-induced changes are summarized in Table 1, where $\perp$ and $/\!/$ express the

**Table 1 | Experimentally determined magnetic moments and hole numbers of FePt.**

| | $m_S \, \mu_B^{-1}$ | $-7m_T \, \mu_B^{-1}$ | $m_L^\perp \, \mu_B^{-1}$ | $m_L^{//} \, \mu_B^{-1}$ | $\frac{\delta m_L^\perp}{m_L^\perp}$ | $\frac{\delta(m_S - 7m_T^\perp)}{m_S - 7m_T^\perp}$ | $\delta n_{5d}$ | $\delta n_{total}$ |
|---|---|---|---|---|---|---|---|---|
| Pt | $0.30 \pm 0.01$ | $< 0.01$ | $0.055 \pm 0.002$ | $0.075 \pm 0.003$ | $(4 \pm 7\%)$ | $13 \pm 3$ | $0.019 \pm 0.001$ | $\sim 0.06$ |
| Fe | $2.39 \pm 0.03$ | $< 0.03$ | $0.13 \pm 0.01$ | $0.10 \pm 0.01$ | | | | |

Here, $m_S$, $m_T$, $m_L$, $\delta n_{5d}$ and $\delta n_{total}$ represent the spin magnetic moment, magnetic dipole moment, orbital magnetic moment, induced holes of Pt in its $5d$ orbitals and total induced holes at Pt–MgO interface, respectively. These physical parameters were obtained by performing the X-ray absorption spectroscopy experiment. The $\delta n_{total}$ is estimated using a capacitance model. $\delta$ denotes the difference in physical parameters between $\mp 200\,V$ ($\mp 0.7\,V\,nm^{-1}$): $\delta x = x(-200\,V) - x(+200\,V)$. $\mp 0.7\,V\,nm^{-1}$ is an electric field in the MgO dielectric.

magnetic moment when the magnetization is oriented perpendicularly and in-plane, respectively. $\delta$ denotes the difference in physical parameters between $-200$ and $+200\,V$: $\delta x = x(-200\,V) - x(+200\,V)$. $\delta n_{total}$ is the estimated induced hole in the total electron orbital at the Pt–MgO interface. The value of $\delta n_{total}$ is estimated using the simple capacitance model[17] (see Supplementary Note 4). For an electric field of $-1.4\,V\,nm^{-1}$ in the MgO dielectric, $\delta n_{total}$ is estimated to be $+0.06$. To characterize the magnetic moments of Fe, we have set the hole number of the $3d$ orbital at $3.73$, according to our first-principles study. Additional experiments to obtain the data presented in Table 1 are found in Supplementary Note 5 and Supplementary Figs 5 and 6. The physical parameters of both Pt and Fe from the XAS/XMCD experiments are the averages of two monatomic layers.

As shown in equation (1), the effective spin magnetic moment is defined as the sum of the spin magnetic moment and the magnetic dipole moment. Generally, in low-symmetry systems like tetragonal crystals, interfaces and thin films, the atomic electron orbital may possess an electric quadrupole. If the atom is also spin polarized, the electric quadrupole induces the anisotropic part of the spin-density distribution, that is, the magnetic dipole moment[45,51,52]. From equation (4), the magnetic dipole moments projected in the directions perpendicular and in-plane to the film are given by $m_T^\perp = -2\mu_B Q_{zz}\langle S_z \rangle e_z/7\hbar$ and $m_T^{//} = \mu_B Q_{zz}\langle S_x \rangle e_x/7\hbar$, respectively. In a system with a relatively large spin–orbit interaction, such as Pt, the anisotropic part of the spin-density distribution is not determined by an electric quadrupole alone; it also is affected by the spin–orbit interaction. However, for a system with a Pt monoatomic layer on Fe, reports[53] state that the magnetization direction dependence of the magnetic dipole in Pt roughly follows equation (4). Further, the spin-density distribution remains almost frozen, independent of the magnetization direction, despite the large spin–orbit interaction in Pt. Therefore, in the following discussion, we focus on the electric-quadrupole-induced part of the magnetic dipole moment. In contrast to the magnetic dipole moment, the spin magnetic moment is insensitive to the magnetization direction[45,51,53]. In our experiment, without voltage application, the effective spin magnetic moment was identical in magnetizations with either perpendicular or in-plane orientation (see Supplementary Note 5 and Supplementary Fig. 5). This result means that $m_T$ is negligible under a zero-bias voltage ($-7m_T < 0.01$), as shown in Table 1.

Under an external voltage, the effective spin magnetic moment shows large magnetization direction dependence (see Supplementary Note 6 and Supplementary Fig. 7). Thus, observations showed significant induction of the magnetic dipole moment in Pt by an external voltage. In Table 1, the relative change in the effective spin magnetic moment is 13%. Rough estimates suggest that variations in the magnetic dipole moment account for more than 70% of this relative change in the effective spin magnetic moment.

**Theoretical study**. Figure 4a shows a section view of our computational model for the first-principles study (see Methods 4). For the in-plane lattice constant of the $L1_0$-FePt, the value of 0.389 nm, which is same as the literature value of the underlayer Pd, has been employed to follow the STEM results. In the computation, the O atom in MgO has been placed above the Pt atom in the [001] direction at the Pt–MgO interface. This configuration is consistent with the STEM results (see Supplementary Note 2 and Supplementary Fig. 3). Figure 4a also depicts the induced charge density. The induced charge density at an electric field of $+0.732\,V\,nm^{-1}$ in the MgO is subtracted from the induced charge density at $-0.732\,V\,nm^{-1}$. Here, an electric field of $\mp 0.732\,V\,nm^{-1}$ induces $\delta n_{total} = \pm 0.03$, where $\delta n_{total}$ is the total number of induced holes in the Pt-1 atom. The blue and red regions indicate the hole accumulations and depletions, respectively. The system shows a VCMA of $-0.28\,pJ\,V^{-1}\,m^{-1}$ (see Supplementary Note 7 and Supplementary Fig. 8). First, clear electric dipole induction can be confirmed in the MgO dielectric. Second, induced holes, shaded in blue in the figure, are localized in Pt-1 atoms that are the atoms nearest to the MgO dielectric. Considerable electric quadrupoles are induced in all magnetic atoms (Pt-1, Fe-1, Pt-2 and Fe-2), although the electric dipole induction is faint. This action should be a unique property of inhomogeneous electric fields in metals. Such electric quadrupole induction couples with the induction of the magnetic dipole moment and changes the MAE, which will be reviewed later. Table 2 shows the physical parameters, obtained from the first-principles study, relevant to the magnetic moments and electronic state of FePt. In this table, $\delta$ denotes the difference in physical parameters between $\mp 0.732\,V\,nm^{-1}$ ($\delta n_{total} = \pm 0.03$): $\delta x = x(-0.732\,V\,nm^{-1}) - x(+0.732\,V\,nm^{-1})$. In a semiquantitative manner, the first-principles study reproduces the experimental results. Some discrepancies found between Tables 1 and 2 can be attributed to the temperature dependence of magnetic moments in an ultrathin ferromagnetic film and/or a 1% strain in the FePt (see Supplementary Note 2). Moreover, orbital polarization might be underestimated because the first-principles calculation does not completely satisfy Hunt's second rule. Table 2 shows the averaged results of Pt-1 and Pt-2 (Fe-1 and Fe-2).

**Discussion**

Then, we qualitatively discuss the origin for the MAE in FePt|MgO system, treating the second-order perturbation of the spin–orbit interaction. Equation (5) shows the perpendicular MAE ($\Delta E$) when the spin–orbit interaction is treated in the second order[45]:

$$\Delta E \cong \frac{\lambda}{4\mu_B}\left(\Delta m_{L,\downarrow} - \Delta m_{L,\uparrow}\right) - \frac{21}{2\mu_B}\frac{\lambda^2}{E_{ex}}\Delta m_T. \qquad (5)$$

The perpendicular MAE is defined as the MAE of the in-plane magnetized film subtracted from the value of the MAE of the perpendicularly magnetized one. Thus, $\Delta m_{L,s}(= m_{L,s}^\perp - m_{L,s}^{//})$ and $\Delta m_T(= m_T^\perp - m_T^{//})$ express the changes in the orbital magnetic

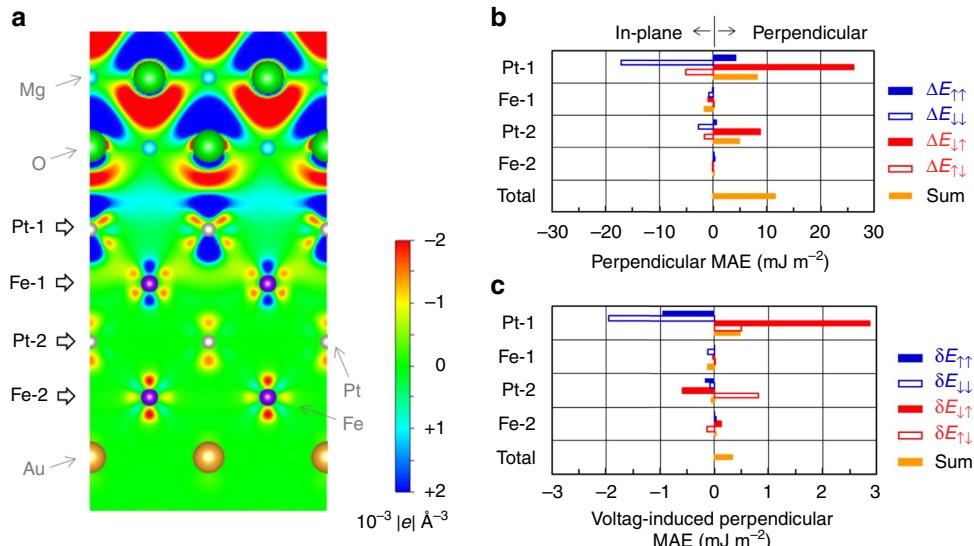

**Figure 4 | Theoretical study.** (**a**) A schematic diagram of the computational model with the induced charge density. The induced charge density at an electric field of $+0.732\,\mathrm{V\,nm^{-1}}$ in the MgO is subtracted from the induced charge density at $-0.732\,\mathrm{V\,nm^{-1}}$. Electric fields of $\mp0.732\,\mathrm{V\,nm^{-1}}$ induce $\delta n_{total} = \pm0.03$, where $\delta n_{total}$ is the total induced holes of the Pt-1 atom. The blue and red areas represent the hole accumulation and depletion, respectively. (**b**) The perpendicular MAE ($\Delta E$) of each monatomic layer calculated with equation (6), where ↑ and ↓ denote the majority and minority spin bands, respectively. (**c**) The voltage-induced perpendicular MAE changes ($\delta E$) calculated with equation (6). The MAE at $+0.732\,\mathrm{V\,nm^{-1}}$ is subtracted from the MAE at $-0.732\,\mathrm{V\,nm^{-1}}$. The spin-flip-term-induced values of MAE ($E_{\downarrow\uparrow}$ and $E_{\downarrow\uparrow}$) of the Pt-1 layer provide the dominant contribution to the perpendicular MAE and its voltage-induced change.

**Table 2 | Calculated magnetic moments and hole numbers of FePt.**

| | $m_S\,\mu_B^{-1}$ | $-7m_T\,\mu_B^{-1}$ | $m_L^{\perp}\,\mu_B^{-1}$ | $m_L^{//}\,\mu_B^{-1}$ | $\frac{\delta m_L^{\perp}}{m_L^{\perp}}$ | $\frac{\delta(m_S-7m_T^{\perp})}{m_S-7m_T^{\perp}}$ | $\delta n_{5d}$ | $\delta n_{total}$ |
|---|---|---|---|---|---|---|---|---|
| Pt | 0.38 | 0.09 | 0.058 | 0.082 | $-1.3\%$ | 2.8% | 0.006 | 0.06 |
| Fe | 2.96 | 0.19 | 0.068 | 0.063 | | | | |

Here, $m_S$, $m_T$, $m_L$, $\delta n_{5d}$ and $\delta n_{total}$ represent the spin magnetic moment, magnetic dipole moment, orbital magnetic moment, induced holes of Pt in its 5d orbitals and total induced holes of Pt-1, respectively. These physical parameters were obtained through a first-principles study. $\delta$ denotes the difference in physical parameters between $\mp0.732\,\mathrm{V\,nm^{-1}}$ ($\delta n_{total} = \pm0.03$): $\delta x = x(-0.732\,\mathrm{V\,nm^{-1}}) - x(+0.732\,\mathrm{V\,nm^{-1}})$. $\mp0.732\,\mathrm{V\,nm^{-1}}$ is an electric field in the MgO dielectric.

moment and the magnetic dipole moment between the perpendicularly ($\perp$) and in-plane ($//$) magnetized electronic states, respectively. Here $\Delta m_{L,\downarrow(\uparrow)}$ expresses the contribution from the minority (majority) spin band. The measured orbital magnetic moment equals to $m_L^{\perp(//)} = m_{L,\downarrow}^{\perp(//)} + m_{L,\uparrow}^{\perp(//)}$. In this case, $\lambda$ is the spin–orbit interaction coefficient. The first term is the perpendicular MAE related to the changes in the orbital magnetic moment. Because the orbital magnetic moment can be derived from the first-order perturbation of the spin–orbit interaction, this term is the second order of the spin–orbit interaction. When the majority spin band is full, $m_{L,\uparrow}^{\perp(//)}$ can be neglected. Then, the perpendicular MAE is proportional to the measured changes in the orbital magnetic moment; this relation is known as Bruno's model[44]. However, $m_{L,\uparrow}^{\perp(//)}$ cannot be neglected in the case of Pt because of the small size of the exchange split (see Supplementary Note 7 and Supplementary Fig. 8). The second term represents the perpendicular MAE related to the changes in magnetic dipole moment[45]. It accounts for the spin-flip excitations between the exchange split ($E_{ex}$) majority and minority spin bands, and should be large in materials with large $\lambda$ and small $E_{ex}$. 5d transition metals, such as Pt, with proximity-induced spin polarization can meet these criteria.

Although equation (5) helps one to understand the underlying physics, it does not apply quantitatively to our system because the approximation it offers is quite rough when $\lambda$ and $E_{ex}$ are comparable. Here, when the perpendicular MAE from the second-order perturbation to the spin–orbit interaction is calculated directly from the first-principles study, a quantitative discussion is feasible[54]. We have employed the following equation as the perpendicular MAE[55].

$$\Delta E_{s's} = \lambda^2 \sum_{o,u} \frac{|\langle u,s',\perp|L_z|o,s,\perp\rangle|^2 - |\langle u,s',//|L_x|o,s,//\rangle|^2}{E_{u,s'} - E_{o,s}}.$$

(6)

For Pt-1 and Pt-2 (Fe-1 and Fe-2) in Fig. 4a, 524.8 meV (52.7 meV), which is the spin–orbit coupling of atoms, is employed for the spin–orbit interaction coefficient ($\lambda$). The total perpendicular MAE from the first-principles study ($13.0\,\mathrm{mJ\,m^{-2}}$) is almost identical to the energy calculated with equation (6) ($11.5\,\mathrm{mJ\,m^{-2}}$). Hence, it is worth discussing the perpendicular MAE with equation (6) using such spin–orbit interaction coefficients. The values of the perpendicular MAE at each monoatomic layer (Pt-1, Fe-1, Pt-2 and Fe-2), arising from the spin-conserved term ($s's = \uparrow\uparrow$ or $s's = \downarrow\downarrow$) and spin-flip term ($s's = \downarrow\uparrow$ or $s's = \uparrow\downarrow$), are derived as shown in Fig. 4b. In equation (5), $m_{L,s}^{\perp(//)}$ consists of only spin-conserved terms and

$m_T^{\perp(//)}$ consists of both spin-conserved and spin-flip terms. However, these magnetization direction dependences ($\Delta m_{L,s}$ and $\Delta m_T$) have different properties. As the spin-conserved terms in $m_T^{\perp(//)}$ do not possess orbital angular momenta, they are independent of magnetization direction. Hence, $\Delta m_T$ consists of only spin-flip terms. Thus, the following two clear relations are concluded. First, the perpendicular MAE from the spin-conserved terms ($\Delta E_{\uparrow\uparrow}$ and $\Delta E_{\downarrow\downarrow}$) in equation (6) corresponds to the first terms of equation (5): $\Delta m_{L,\downarrow} - \Delta m_{L,\uparrow}$. Second, the perpendicular MAE from the spin-flip terms ($\Delta E_{\downarrow\uparrow}$ and $\Delta E_{\uparrow\downarrow}$) in equation (6) corresponds to the second term of equation (5): $\Delta m_T$. As shown in Fig. 4b, Pt-1, placed closest to the MgO dielectric material, provides the dominant contribution to the perpendicular MAE in the FePt|MgO system. Note that the anisotropy in the orbital magnetic moment, $\Delta m_{L,\downarrow} - \Delta m_{L,\uparrow}$, in Pt-1 decreases the perpendicular MAE ($\Delta E_{\uparrow\uparrow} + \Delta E_{\downarrow\downarrow}$ in Fig. 4b), which is behaviour contrary to systems with $3d$ transition metals[44]. In contrast, the magnetic dipole moment, $\Delta m_T$, of Pt-1 increases the perpendicular MAE ($\Delta E_{\downarrow\uparrow} + \Delta E_{\uparrow\downarrow}$ in Fig. 4b). This result reveals the important role of the magnetic dipole moment, which comes mainly from the electric quadrupole, in the magnetic anisotropy of ordered alloys.

Finally, the origin of the voltage-induced perpendicular MAE change in our FePt|MgO system is discussed. The MAE change, defined as $\delta E = \Delta E(-0.732\,\mathrm{V\,nm}^{-1}) - \Delta E(+0.732\,\mathrm{V\,nm}^{-1})$ at each monoatomic layer, is displayed in Fig. 4c. In a manner similar to the one exhibited by the perpendicular MAE in Fig. 4b, Pt-1 also supplies the dominant contribution to the MAE change in the FePt|MgO system. We found, first, that the voltage induction of the anisotropy in the orbital magnetic moment of Pt-1 decreases the perpendicular MAE ($\delta E_{\uparrow\uparrow} + \delta E_{\downarrow\downarrow}$ in Fig. 4c). The sign of the VCMA from this mechanism is opposite to the experimentally observed VCMA with $3d$ transition metals, such as Fe(Co)|MgO (refs 17,20,23–27). Second, the voltage induction of the magnetic dipole moment of Pt-1 increases the perpendicular MAE ($\delta E_{\downarrow\uparrow} + \delta E_{\uparrow\downarrow}$ in Fig. 4c). The increase in the perpendicular MAE by the induction of the magnetic dipole moment is greater than the decrease by the induction of the anisotropy in the orbital magnetic moment. Then, the total perpendicular MAE in the FePt|MgO system increases under the condition of electron depletion at the Pt–MgO interface, which is consistent with situation depicted in Fig. 1d. Because of the mechanism by which the magnetic dipole moment contributes, the FePt|MgO system shows the VCMA with the same polarity reported in Fe(Co)|MgO systems[17,20,23–27].

We know of two mechanisms that account for voltage-induced perpendicular MAE changes. The first mechanism comes from charge doping inducing anisotropy in the orbital magnetic moment. Because each electron orbital in the vicinity of the Fermi level has a different density of state, selective charge doping may induce anisotropy in the orbital magnetic moment. This effect changes the perpendicular MAE[44,45,51], as expressed in the first term of equation (5). The present study focuses on the second mechanism, that is, the perpendicular MAE change from the induction of a magnetic dipole moment. Because an electric field applied to the metal–dielectric is inhomogeneous because of the strong electrostatic screening effect in metals, such an electric field, including higher-order quadratic components, can couple with the electric quadrupole correlating to the magnetic dipole moment in an electron shell in a metal layer as shown in Fig. 1a (inset). The induced energy split of each orbital changes the magnetic anisotropy through spin-flip excitation[45], as shown in the second term in equation (5). In our FePt|MgO system, the VCMAs from the orbital magnetic moment and the magnetic

dipole moment partially cancel out one another. It should be noted that the VCMA from the mechanism is larger than the VCMA of the orbital magnetic moment. However, total VCMA is only 9% of the VCMA from the magnetic dipole moment. If a material with reducing the VCMAs canceling out is to be designed, an electronic VCMA larger by more than a factor of 10 would be feasible. In ordered alloys, it is reported that both the orbital magnetic moment and magnetic dipole moment can be modified by controlling a chemical order[56]. Hence, it is expected that the VCMA mechanism is also controlled in the same manner. A material with reducing the VCMAs canceling out should be found in such variety kinds of ordered alloy.

To evaluate the robustness of the present study, a controlled experiment with (0 0 1)-oriented epitaxial multilayers of Fe(0.5 nm)|Pt|MgO was conducted. (see Supplementary Note 1 and Supplementary Fig. 2). First, the VCMA was sensitive to the Pt layer thickness and was almost maximized when one Pt monatomic layer (0.2 nm) was employed. Second, an induced XMCD similar to that shown in Fig. 2 was also observed, and it was scaled with an external electric field and a number of Pt monatomic layers. Hence, the one monatomic layer of Pt at the Fe–MgO interface contributes the most to the enhanced VCMA in the system. This is consistent with the first-principles study shown in Fig. 4c. In our chemically ordered FePt alloy, one monatomic layer of coverage by the Pt was the best condition for VCMA; however, less than one monatomic layer of coverage may exhibit their VCMA maximum in the case of chemically disordered alloys, which should show different orbital and dipole contributions[56].

Our experimental results (see Table 1) indicate that the voltage-induced change in the hole number of the Pt 5d orbital is 0.019. If all induced charge is doped in the $d$ orbital with the largest electric quadrupole (such as $Q_{zz} = -4/7$ for the $d_{3r^2-z^2}$ orbital), such doping will result in an increase of $Q_{zz}$ of $\sim 0.01$. In contrast, our experimental result produced a rough estimation indicating that a $Q_{zz}$ of $\sim 0.06$ was induced. This experimental result, a sixfold increase in the electric quadrupole, cannot be explained only by charge doping. Therefore, it provides evidence that the metal–dielectric interface is a unique system in which the electric quadrupole in an atom can be modulated intensively by redistributing the orbital occupancy under the application of an external electric field. It should be noted that the quadratic electric field potential induced by the inhomogeneous electric field at the metal–dielectric interface plays an important role, as shown in the inset of Fig. 1a. This large quadrupole induction in metal may be correlated with the recent reports of electric-field-induced ferromagnetic phase transitions in Pt[12] and magnetic moment induction in Pd[37,41]. Further, the demonstrated large quadrupole induction in metal may show an interplay with superconductivity because discussions about non-BCS (Bardeen, Cooper and Schrieffer) superconducting mechanisms have focused on the role of multipoles[57].

## Methods

**Sample preparation.** A multilayer of MgO(0 0 1)substrate|MgO buffer (3 nm)|Cr(20 nm)|Pd(50 nm)|Fe(0.14 nm)|Pt(0.20 nm|Fe(0.14 nm)|Pt(0.20 nm)|MgO barrier (5 nm) was fabricated by molecular-beam epitaxy under ultra-high vacuum. The FePt had an $L1_0$ structure, which was confirmed by reflection high-energy electron diffraction spectroscopy. During deposition of the MgO buffer, Cr and FePt, the substrate temperature was kept at 260 °C. The Pd layer was deposited at room temperature and was postannealed at 300 °C. Immediately upon removal from the vacuum, the film was spin coated with a polyimide (1,500 nm) and annealed at 200 °C. A top electrode (Cr(20 nm)|Pd(10 nm)) was prepared by electron-beam deposition using a shadow-mask method. The designed junction was 1 mm in diameter. The top electrodes were connected to the voltage supply using aluminium wires with indium pastes.

**Scanning transmission electron microscopy.** High-angle annular dark-field scanning transmission electron microscopy observations were performed using a scanning transmission electron microscope (FEI Titan G2 80–200) with a probe aberration corrector. The specimen for STEM observation was prepared by the lift-out technique using a FEI Helios Nanolab 650.

**X-ray absorption spectroscopy.** XAS/XMCD measurements at the Pt $L$ edges with external voltage applications were performed at BL39XU at the SPring-8 synchrotron radiation facility. Monochromatic and circularly polarized X-rays ($P_c > 95\%$) were produced using a Si 111 double-crystal monochromator and a diamond X-ray phase retarder. The helicity-dependent X-ray absorption spectrum was determined by monitoring the X-ray fluorescence yield from the sample as the photon helicity was reversed at 0.5 Hz, while the X-ray energy was scanned around the Pt $L_3$ and $L_2$ edges. X-ray fluorescence yields were measured using a four-element silicon drift detector (Bruker D461-4X10), which is operational with an input count rate of X-ray photons as high as $2 \times 10^5$ counts s$^{-1}$ and has an energy resolution of 240 eV. The X-ray beam-spot size was ~0.1 mm. Spectra of an FePt film were recorded in an external magnetic field parallel to the X-ray propagation direction as shown in Fig. 1b. Ar gas was blown onto the sample during the measurements to reduce time-dependent changes in the FePt magnetic properties (that is, degradation induced by irradiation by an intense X-ray beam). The polarization-averaged XAS spectrum was defined by $(\mu_+ + \mu_-)/2$, where $\mu_+$ and $\mu_-$ were the spectra taken with right and left helicities, respectively. The XMCD spectrum was determined from the difference between the two helicity spectra, that is, $\mu_+ - \mu_-$. Instrumental asymmetries or non-magnetic backgrounds were cancelled by averaging the spectra measured for the reversed magnetization directions. The details of the XMCD measurements are discussed elsewhere[52]. All measurements were conducted at room temperature and atmosphere. The XAS spectra were normalized by taking into account the correction ratio. A ratio 2.22, the ratio of the $L_3$ to the $L_2$, was employed[58].

XAS/XMCD measurements at the Fe $L$ edges were carried out at BL25SU at SPring-8. Total electron yield (TEY) methods were employed, and the TEY current at the sample was systematically normalized by the incoming beam intensity, monitored by measuring the TEY on the surface of a postfocusing mirror.

**First-principles study.** First-principles electronic structure calculations were carried out using the projector augmented-wave method, implemented in the Vienna *ab initio* simulation package[59]. For the exchange-correlation energy, we used a generalized gradient approximation given by Perdew et al.[60] All calculations were performed with a $24 \times 24 \times 1$ Monkhorst-Pack k-point mesh and a 500 eV plane-wave cutoff energy. The magnetic force theorem was used to estimate the MAE. The FePt|MgO film was modelled by the periodic slab supercell with 7 monolayer (ML) Au, 4 ML FePt, 5 ML MgO and an 18-Å-thick vacuum layer. The external electric field was introduced by using the dipole layer method[61]. The atomic coordinates in the slab were fully relaxed until the atomic forces were $<0.02$ eV Å$^{-1}$ by keeping the in-plane lattice constant fixed to the same value, 0.389 nm, as the Pd underlayer in the experiment. The atomic-sphere radii used to evaluate the atomically resolved magnetic moments and the values of MAE from second-order perturbation formulae was 1.5, 1.4, 1.5, 1.5 and 0.9 Å for Au, Fe, Pt, O and Mg atoms, respectively.

**Data availability.** The data that support these findings are available from the corresponding author (S.M.) on reasonable request.

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

## Acknowledgements

We thank Y. Shiratsuchi and R. Nakatani of Osaka University for assistance with the vibrating-sample magnetometry measurements. We also thank R. Miyakaze, T. Kawabe, K. Shimose and T. Furuta of Osaka University for assistance with the XAS/XMCD measurements. This work received support from the ImPACT Program of the Council for Science, Technology and Innovation (Cabinet Office, Government of Japan) and from JSPS KAKENHI (Nos JP26103002, JP15H05420), $MI^2I$ program and the Murata Science Foundation. The XAS and XMCD measurements were performed in SPring-8 with the approval of the Japan Synchrotron Radiation Research Institute (Proposal Nos 2014B1007, 2015A1003, 2015A1572, 2015B0079, 2015B1020, 2015B1250, 2016A0079, 2016A1122, 2016A1314 and 2016B1017).

## Author contributions

S.M., M.Su., K.M., K.T. and T.T conducted the XAS/XMCD experiment of Pt. M.T. and M.Sh. conducted the first-principles study. T.N. and S.Y. prepared the FePt multilayer. S.M., K.M., T.T. and K.N. conducted the device fabrication. M.G., Y.K. and T.N. conducted the XAS/XMCD experiment of Fe. T.O. and K.H. conducted the STEM characterization. F.B., E.T. and Y.S. developed the explanation of the experiment. M.Su. and Y.S. supervised the project. S.M. conducted the analysis and wrote the manuscript. All authors discussed the results and commented on the manuscript.

## Additional information

**Competing interests:** The authors declare no competing financial interests.

