## [Peer Review File · Nature Communications]

Reviewers' Comments:

Reviewer #1 (Remarks to the Author):

Please see attached file.

Reviewer #2 (Remarks to the Author):

This is an interesting and timely paper, showing a new mechanism of VCMA based on voltage-induced induction of magnetic dipole moment in Pt, rather than voltage-induced change of orbital magnetic anisotropy energy, which is the mechanism mostly studied in previous works. Understanding of the VCMA microscopic mechanisms is important to the development of electric-field-controlled magnetic memory devices, and hence I think this study can have a large impact on the spintronics and magnetic memory community. I do have several questions and suggestions:

- Use of the term Shannon limit to describe the lowest possible switching energy should be explained. Normally this refers to the rate of data communication in a noisy channel. Do the authors actually refer to the Landauer limit?
- What are the design guidelines to realize this new VCMA mechanism in new material structures? What other materials (other than Pt) might be expected to show this effect? The authors mention that if a material with both VCMA mechanisms having the same sign were to be designed, more than ten times larger electronic VCMA would be feasible. The paper should discuss what materials might exhibit such as effect, in particular given that this could be calculated from first principles.
- For applications, the biggest question will be how material specific this effect is. The L10 structure used in this paper is not really suitable for MRAM given the low TMR values which would be expected in a magnetic tunnel junction built from this material. For example if the authors used 3 monolayers of Fe | 1 monolayer of Pt, or if the structure was 001 instead of 111, would they see a similar effect?
- On a related note, how sensitive is this effect to Pt thickness? Can a dusting layer with nominal

thickness less than one monolayer accomplish the same goal?

- In general, I think the latter two questions illustrate a need for control experiments with varying thicknesses and materials, to evaluate the robustness and practical application potential of this effect.

- Can the authors rule out electrochemical effects, given the slow measurement method and the very large voltages applied?

- It would be helpful to applied researchers and device physicists if the units of VCMA were put in the usually used ones, such as anisotropy energy change per electric field. Also, it may help to move some of the information from Supplementary section I to the main manuscript text.

- Occasional typos and grammatical errors should be fixed.

Reviewer #3 (Remarks to the Author):

Voltage controlled magnetic anisotropy (VCMA) holds the key to the next generation ultra-low energy spintronics devices. The authors discovered a new mechanism for VCMA in FePt/MgO structure. In addition to the conventional contribution to VCMA due to voltage induced modification of orbital moments, the author identified a new source that is related to the magnetic dipole moment produced by the electric quadrupole. The change of magnetic anisotropy energy (MAE) can be larger with the dipole effect, which could lead to a potential large enhance of the VCMA effect. This work is novel, important and will be interesting a wide range of readers. Therefore this study is suitable for a premium journal like Nature Communications, if the authors can address my following questions:

1. This work highlights the importance of Pt in obtaining the large contribution to VCMA as described by the second term of Eq. 5. However it is know if Pt is directly placed next to MgO, the tunneling magnetoresistance (TMR) of the junction will be diminished. Would this quadrupole induced effect also exist in normal FM such as Fe, Co and Ni? Because if it only exists with heavy metals with large spin orbit interaction, it will inevitably destroy the TMR therefore making this VCMA mechanism less useful.

2. It is discovered the VCMA effects due to orbital moment and induced magnetic dipole moment cancel each other. It is speculated that the VCMA efficiency can be dramatically improved if the signs of the two effects are the same. Can the authors discuss under what situations we might expect the signs to be the same?

3. What is the VCMA effect in the present study in terms of pJ/Vm?

4. What is the uncertainty of Δ_n total in the table 1?
5. The format of citations (line 57-63) is not consistent.

Comments from Reviewer 1:

Miwa, *et al* in their manuscript NCOMMS-16-27105-T “*Voltage control of platinum orbits: a contribution to interfacial magnetism*” carried out complementary experimental and theoretical studies of the voltage controlled magnetic anisotropy (VCMA) of Pt/MgO interfaces in FePt/MgO systems. The VCMA has attracted a lot of attention recently because it has great potential in ultralow-power magnetoelectric random access memory (MeRAM) devices. Currently, the proposed underlying mechanism responsible for the VCMA is the selective electric-field induced electron redistribution of interfacial magnetic atoms between different orbitals. One of the bottlenecks is the low VCMA efficiency (about 30-100 fJ/(Vm)). The authors here propose a new mechanism to enhance the electronic VCMA based on the electric quadrupole which, if the atom is spin-polarized, induces a magnetic dipole moment. The ab initio calculations reveal that the response of the magnetic dipole moment to electric field is dominant. In addition the experimental results give strong evidence of a large increase electric-field induced increase in the electric quadrupole moment.

In particular I find interesting the proposal to search for materials where the contributions of the orbital magnetic moment and the magnetic dipole moment have the same sign which can in turn lead to large VCMA coefficient.

The results are interesting and the proposed underlying mechanism of the electric quadrupole moment is novel. This work will hopefully stimulate more experimental and theoretical research and has the potential to increase the VCMA efficiency more than ten times. Consequently, the manuscript does warrant publication in Nature Communications after the authors respond to the following comments and revise the manuscript accordingly.

1. There is a large discrepancy of the spin magnetic moment, m_s , of the Fe atom between experiment (Table 1) and theory (Table 2). The experimental value is $2.39 \mu_B$ while the theoretical value is $2.96 \mu_B$. The authors should comment on this. Perhaps this is due to inherent biaxial strain between FePt and MgO, where the former has a lattice constant of 0.389 nm while the latter has a lattice constant of 0.42 nm.
2. Similarly, there is large discrepancy (about a factor of two) of the out-of-plane and in-plane orbital moment between experiment (Table 1) and theory (Table 2). Why?
3. Bruno’s expression Eq. (5) is valid for elemental solids. However, in the case of two-atom binary system (FePt) the validity of this expression is questionable. Which spin-orbit coupling should one use? That of Fe or Pt? The authors should justify the use of Eq. (5) and explain more details.
4. The following requires clarification and more discussion. At the bottom of page 10 the authors state “*In Eq. 5, $m_{L,S}^{\perp(\parallel)}$ consists **only** of spin-conserved terms and $m_T^{\perp(\parallel)}$ consists of **both** spin-conserved and spin-flip terms.*” This is consistent with Ref. 36. However, the next statement is unclear and inconsistent with the previous statement. Namely, the authors state “*First, the perpendicular MAE from the spin conserved terms ($\Delta E_{\uparrow\uparrow}$ and $\Delta E_{\downarrow\downarrow}$) corresponds to the first term of Eq. 5, $\Delta m_{L\downarrow} - \Delta m_{L\uparrow}$. Second the perpendicular MAE from the spin-flip terms ($\Delta E_{\uparrow\downarrow}$ and $\Delta E_{\downarrow\uparrow}$) corresponds to the second term of Eq. 5.*” The latter sentence is inconsistent with the former. The authors should clarify more this important point.
5. The authors should quote what is the ab initio value of VCMA coefficient in pJ/(Vm)?

6. Why is the underlying mechanism of the large value of the electric quadrupole moment, $Q_{zz} = 0.06$ (bottom of page 12)?
7. In Figure S8 the authors plot various quantities versus the change of total number electrons, Δn_{TOT} . Is this the change of the total number of electrons on the Pt interfacial atom or change of total number of electrons in the unit cell? It will be instructive to show also the electric field in the insulator (upper horizontal scale) since the input of VASP is the external electric field in vacuum? This will help the reader to see what is the E-field that gives rise to a specific change of Δn_{TOT} .
8. The authors should add a recent relevant publication on the giant VCMA in Au/FeCo/MgO “*Electric-field-driven magnetization switching and nonlinear magnetoelasticity in Au/FeCo/MgO heterostructures*” P. V. Ong, Nicholas Kioussis, P. Khalili Amiri, K. L. Wang, Scientific Reports 6, Article number: 29815 (2016), doi:10.1038/srep29815.

Response to Reviewer #1

We would like to thank Reviewer #1 for his/her precise review of our manuscript as well as a high evaluation. The suggestions based on the theoretical viewpoint were very useful for the revision. We have revised the manuscript as follows.

Miwa, *et al* in their manuscript NCOMMS-16-27105-T “*Voltage control of platinum orbits: a contribution to interfacial magnetism*” carried out complementary experimental and theoretical studies of the voltage controlled magnetic anisotropy (VCMA) of Pt/MgO interfaces in FePt/MgO systems. The VCMA has attracted a lot of attention recently because it has great potential in ultralow-power magnetoelectric random access memory (MeRAM) devices. Currently, the proposed underlying mechanism responsible for the VCMA is the selective electric-field induced electron redistribution of interfacial magnetic atoms between different orbitals. One of the bottlenecks is the low VCMA efficiency (about 30-100 fJ/(Vm)). The authors here propose a new mechanism to enhance the electronic VCMA based on the electric quadrupole which, if the atom is spin-polarized, induces a magnetic dipole moment. The *ab initio* calculations reveal that the response of the magnetic dipole moment to electric field is dominant. In addition the experimental results give strong evidence of a large increase electric-field induced increase in the electric quadrupole moment.

In particular I find interesting the proposal to search for materials where the contributions of the orbital magnetic moment and the magnetic dipole moment have the same sign which can in turn lead to large VCMA coefficient.

The results are interesting and the proposed underlying mechanism of the electric quadrupole moment is novel. This work will hopefully stimulate more experimental and theoretical research and has the potential to increase the VCMA efficiency more than ten times. Consequently, the manuscript does warrant publication in Nature Communications after the authors respond to the following comments and revise the manuscript accordingly.

1. There is a large discrepancy of the spin magnetic moment, m_s , of the Fe atom between experiment (Table 1) and theory (Table 2). The experimental value is 2.39 μ_B while the theoretical value is 2.96 μ_B . The authors should comment on this. Perhaps this is due to inherent biaxial strain between FePt and MgO, where the former has a lattice constant of 0.389 nm while the latter has a lattice constant of 0.42 nm.

2. Similarly, there is large discrepancy (about a factor of two) of the out-of-plane and in-plane orbital moment between experiment (Table 1) and theory (Table 2). Why?

We characterized the in-plane lattice constant of our FePt film by STEM, as discussed in Supplementary Information 2. While the literature value of the in-plane lattice constant is 0.385 nm, the lattice constant of our FePt is 0.389 nm. As the reviewer mentioned, such a 1%-strain in the FePt film may induce the discrepancy. Moreover, from STEM, there is one lattice dislocation of dozens of atoms at the FePt|MgO interface. This dislocation is probably attributable to an 8% lattice mismatch between the literature values of $L1_0$ -FePt (0.385 nm) and MgO (0.421 nm). This implies that the strain in the FePt is not very large as expected from a large difference of lattice constants between FePt and MgO.

An alternative explanation is the temperature dependence of magnetic moments. Ultrathin ferromagnetic films are known to exhibit larger temperature dependence than bulks. In Tables 1 and 2, the spin magnetic moments measured at room temperature are about 20% smaller than that of the theory at 0 K. The discrepancy can be attributed to the temperature dependence. In addition, as the first-principles calculation does not completely satisfy Hunt's second rule, orbital polarization might be underestimated.

We would like to modify our manuscript as follows. Thank you for the suggestions.

REVISION 1

(P10L235, *before* revision)

The first-principles study, in a semiquantitative fashion, reproduces the experimental results.

(P10L235, *after* revision)

In a semiquantitative fashion, the first-principles study reproduces the experimental results. Some discrepancies found between Tables 1 and 2 can be attributed to the temperature dependence of magnetic moments in an ultrathin ferromagnetic film and/or a 1%-strain in the FePt (see Supplementary Information 2). Moreover, orbital polarization might be underestimated because the first-principles calculation does not completely satisfy Hunt's second rule.

3. Bruno's expression Eq. (5) is valid for elemental solids. However, in the case of two-atom binary system (FePt) the validity of this expression is questionable. Which spin-orbit coupling should one use? That of Fe or Pt? The authors should justify the use of Eq. (5) and explain more details.

We employed spin-orbit coefficients (λ) of 524.8 and 52.7 meV for Pt and Fe, respectively. These values are the spin-orbit couplings of each atom. The magnetic anisotropy energy from the first-principles study (13.0 mJ m⁻²) is almost identical to the energy calculated from the second-order perturbation (11.5 mJ m⁻²) using $\Delta E_{s's} = \lambda^2 \sum_{o,u} \left(\left| \langle u, s', \perp | L_z | o, s, \perp \rangle \right|^2 - \left| \langle u, s', // | L_x | o, s, // \rangle \right|^2 \right) / (E_{u,s'} - E_{o,s})$. Hence, it is worth discussing the use of the equation above with such spin-orbit interaction coefficients. We would like to add an explanation to the main text. Thank you for the comment.

REVISION 2

(P11L264, *before* revision)

Although Eq. 5 helps one to understand the underlying physics, it does not apply quantitatively to our system because the approximation it offers is quite rough when λ and E_{ex} are comparable. Here, when the perpendicular MAE from the second-order perturbation to the spin-orbit interaction is calculated directly from the first-principles study, a quantitative discussion is feasible^{Error! Reference source not found.}. We have employed

$$\Delta E_{s's} = \lambda^2 \sum_{o,u} \left(\left| \langle u, s', \perp | L_z | o, s, \perp \rangle \right|^2 - \left| \langle u, s', // | L_x | o, s, // \rangle \right|^2 \right) / (E_{u,s'} - E_{o,s}) \quad \text{as the}$$

perpendicular MAE^{Error! Reference source not found.}.

(P11L264, *after* revision)

Although Eq. 5 helps one to understand the underlying physics, it does not apply quantitatively to our system because the approximation it offers is quite rough when λ and E_{ex} are comparable. Here, when the perpendicular MAE from the second-order perturbation to the spin-orbit interaction is calculated directly from the first-principles study, a quantitative discussion is feasible⁵³. We have employed the following equation as the perpendicular MAE⁵⁴.

$$\Delta E_{s's} = \lambda^2 \sum_{o,u} \frac{\left| \langle u, s', \perp | L_z | o, s, \perp \rangle \right|^2 - \left| \langle u, s', // | L_x | o, s, // \rangle \right|^2}{E_{u,s'} - E_{o,s}} \quad (6)$$

For Pt-1 and Pt-2 (Fe-1 and Fe-2) in Fig. 4a, 524.8 meV (52.7 meV), which is the spin-orbit coupling of atoms, is employed for the spin-orbit interaction coefficient (λ). The total perpendicular MAE from the first-principles study (13.0 mJ m⁻²) is almost identical to the energy calculated with Eq. 6 (11.5 mJ m⁻²). Hence, it is worth discussing the perpendicular MAE with Eq. 6 using such spin-orbit interaction

coefficients.

4. The following requires clarification and more discussion. At the bottom of page 10 the authors state “In Eq. 5, $m_L \perp(\parallel)$ consists **only** of spin-conserved terms and $m_T \perp(\parallel)$ consists of **both** spin-conserved and spin-flip terms.” This is consistent with Ref. 36. However, the next statement is unclear and inconsistent with the previous statement. Namely, the authors state “First, the perpendicular MAE from the spin conserved terms ($\Delta E_{\uparrow\uparrow}$ and $\Delta E_{\downarrow\downarrow}$) corresponds to the first term of Eq. 5, $\Delta m_{L\downarrow} - \Delta m_{L\uparrow}$. Second the perpendicular MAE from the spin-flip terms ($\Delta E_{\uparrow\downarrow}$ and $\Delta E_{\downarrow\uparrow}$) corresponds to the second term of Eq. 5.” The latter sentence is inconsistent with the former. The authors should clarify more this important point.

As we mentioned in the main text, m_L consists of only spin-conserved terms and m_T consists of both the spin-conserved and spin-flip terms. However, their angular dependences are different. As the spin-conserved term in the m_T does not have an angular dependence because of its orbital angular momentum of $L_z = 0$, the angular-dependent components of m_T ($= \Delta m_T$) only consists of spin-flip terms. We would like to include additional explanations in the main text. Thank you for the suggestion.

REVISION 3

(P11L276, before revision)

In Eq. 5, $m_{L,s}^{\perp(\parallel)}$ consists only of spin-conserved terms and $m_T^{\perp(\parallel)}$ consists of both spin-conserved and spin-flip terms. However, these magnetisation direction dependences ($\Delta m_{L,s}$ and Δm_T) exhibit two clear relations³⁶. First, the perpendicular MAE from the spin-conserved terms ($\Delta E_{\uparrow\uparrow}$ and $\Delta E_{\downarrow\downarrow}$) corresponds to the first terms of Eq. 5: $\Delta m_{L,\downarrow} - \Delta m_{L,\uparrow}$. Second, the perpendicular MAE from the spin-flip terms ($\Delta E_{\downarrow\uparrow}$ and $\Delta E_{\uparrow\downarrow}$) corresponds to the second term of Eq. 5: Δm_T .

(P11L276, after revision)

In Eq. 5, $m_{L,s}^{\perp(\parallel)}$ consists of only spin-conserved terms and $m_T^{\perp(\parallel)}$ consists of both spin-conserved and spin-flip terms. However, these magnetisation direction dependences ($\Delta m_{L,s}$ and Δm_T) have different properties. As the spin-conserved terms in

$m_T^{\perp(l)}$ do not possess orbital angular momenta, they are independent of magnetisation direction. Hence, Δm_T consists of only spin-flip terms. Thus, the following two clear relations are concluded. First, the perpendicular MAE from the spin-conserved terms ($\Delta E_{\uparrow\uparrow}$ and $\Delta E_{\downarrow\downarrow}$) in Eq. 6 corresponds to the first terms of Eq. 5: $\Delta m_{L,\downarrow} - \Delta m_{L,\uparrow}$. Second, the perpendicular MAE from the spin-flip terms ($\Delta E_{\downarrow\uparrow}$ and $\Delta E_{\uparrow\downarrow}$) in Eq. 6 corresponds to the second term of Eq. 5: Δm_T .

5. The authors should quote what is the ab initio value of VCMA coefficient in pJ/(Vm)?

The VCMA coefficient in the ab initio study was $0.28 \text{ pJ V}^{-1} \text{ m}^{-1}$. We modify the main text as follows.

REVISION 4

(P09L225, *after* revision)

The system shows a VCMA of $-0.28 \text{ pJ V}^{-1} \text{ m}^{-1}$. (See Supplementary Information 6.)

REVISION 5

(SI, P10L166, *after* revision)

The VCMA coefficient in the system is $-0.28 \text{ pJ V}^{-1} \text{ m}^{-1}$.

6. Why is the underlying mechanism of the large valued of the electric quadrupole moment, $Q_{zz} = 0.06$ (bottom of page 12)?

Thank you for the comment. As discussed in the main text, such a large $Q_{zz} = 0.06$ cannot be explained by charge doping to the $5d$ -orbital. Hence, it should be induced by charge redistribution in $5d$ -orbitals. Such a charge redistribution can be caused by an induced quadratic electric field potential due to *inhomogeneous* electric fields at the metal|dielectric interface, as shown in Fig. 1a. We would like to include additional comments in our manuscript.

REVISION 6

(P14L339, *before* revision)

In contrast, our experimental result produced a rough estimation showing that a value of Q_{zz} of approximately 0.06 was induced. This experimental result, a six-fold increase in

the electric quadrupole moment, gives evidence that the metal|dielectric interface is a unique system in which the electric quadrupole in an atom can be modulated intensively by redistributing the orbital occupancy under the application of an external electric field.

(P14L339, *after* revision)

In contrast, our experimental result produced a rough estimation indicating that a Q_{zz} of approximately 0.06 was induced. This experimental result, a six-fold increase in the electric quadrupole moment, cannot be explained only by charge doping. Therefore, it provides evidence that the metal|dielectric interface is a unique system in which the electric quadrupole in an atom can be modulated intensively by redistributing the orbital occupancy under the application of an external electric field. It should be noted that the quadratic electric field potential induced by the inhomogeneous electric field at the metal|dielectric interface plays an important role, as shown in the inset of Fig. 1a.

7. In Figure S8 the authors plot various quantities versus the change of total number electrons, Δn_{TOT} . Is this the change of the total number of electrons on the Pt interfacial atom or change of total number of electrons in the unit cell? It will instructive to show also the electric field in the insulator (upper horizontal scale) since the input of VASP is the external electric field in vacuum? This will help the reader to see what is the E-field that gives rise to a specific change of Δn_{TOT} .

Thanks for the suggestion. As the reviewer said, δn_{total} is the change in the total number of holes on the Pt interfacial atom (Pt-1 in Fig. 4a). We would like to add a description of the electric field in the MgO insulator. We modified our manuscript as follows.

REVISION 7

(P09L221, *before* revision)

Figure 4a depicts the outcome obtained from the induced charge density at $\delta n_{total} = -0.03$ subtracted from the induced charge density at $\Delta n_{total} = +0.03$.

(P09L221, *after* revision)

Figure 4a also depicts the induced charge density. The induced charge density at an electric field of $+0.732 \text{ V nm}^{-1}$ in the MgO is subtracted from the induced charge density at -0.732 V nm^{-1} . Here, an electric field of $\mp 0.732 \text{ V nm}^{-1}$ induces $\delta n_{total} = \pm 0.03$, where δn_{total} is the total number of induced holes in the Pt-1 atom.

REVISION 8

(Supplementary Information: P11L179, *before* revision)

(Supplementary Information: P11L179 *after* revision)

8. The authors should add a recent relevant publication on the giant VCMA in Au/FeCo/MgO “*Electric-field-driven magnetization switching and nonlinear magnetoelasticity in Au/FeCo/MgO heterostructures*” P. V. Ong, Nicholas Kioussis, P. Khalili Amiri, K. L. Wang, Scientific Reports 6, Article number: 29815 (2016), doi:10.1038/srep29815.

Thank you for informing us the interesting and important paper. We included the paper as Ref. 42 of the revised manuscript. We carefully checked the relevant papers again and revised the reference. (add Refs. 20, 30, 31, 37, 39, 40, and 48)

Response to Reviewer #2

We would like to thank Reviewer #2 for his/her precise review of our manuscript. We also appreciate the reviewer for recognizing the importance of our work. The number of valuable comments were very useful for the revision. We have revised the manuscript as follows.

This is an interesting and timely paper, showing a new mechanism of VCMA based on voltage-induced induction of magnetic dipole moment in Pt, rather than voltage-induced change of orbital magnetic anisotropy energy, which is the mechanism mostly studied in previous works. Understanding of the VCMA microscopic mechanisms is important to the development of electric-field-controlled magnetic memory devices, and hence I think this study can have a large impact on the spintronics and magnetic memory community. I do have several questions and suggestions:

- Use of the term Shannon limit to describe the lowest possible switching energy should be explained. Normally this refers to the rate of data communication in a noisy channel. Do the authors actually refer to the Landauer limit?

Thank you for the suggestion. As the reviewer mentioned, the lowest possible switching energy is the Landauer limit, $k_B T \ln 2$. In our original manuscript, we employed $100 k_B T$ as the thermodynamic limit for the GHz logic in Si [J. D. Meindl, Q. Chen, and J. A. Davis, *Science* **293**, 2044–2049 (2001)]. However, it would be much better to mention the Landauer limit in our introduction paragraph. We would like to modify the manuscript as follows.

REVISION 9

(P2L41, *before* revision)

The energy required for magnetisation switching is more than 10^5 times the size of the Shannon limit of approximately $100 k_B T$, where k_B is the Boltzmann constant.

(P2L41, *before* revision)

The energy required for magnetisation switching is more than 10^7 times the Landauer limit of $k_B T \ln 2$, where k_B is the Boltzmann constant.⁷

- What are the design guidelines to realize this new VCMA mechanism in new material structures? What other materials (other than Pt) might be expected to show this effect? The authors mention that if a material with both VCMA mechanisms having the same sign were to be designed, more than ten times larger electronic VCMA would be feasible. The paper should discuss what materials might exhibit such an effect, in particular given that this could be calculated from first principles.

Thank you for the valuable comment. As discussed in ref. [P. Kamp *et al.*, *Phys. Rev. B* **59**, 1105 (1999)], the orbital magnetic moment and magnetic dipole moment can be

modulated by controlling a chemical order. Hence, it is expected that the VCMA mechanism can also be controlled in the same manner. A material with both VCMA mechanism adding up should be found in such variety kinds of ordered alloy.

REVISION 10

(P13L320, *before* revision)

If a material with both VCMA in the same polarity are to be designed, more than ten times larger electronic VCMA would be feasible.

(P13L320, *after* revision)

If a material with both VCMA in the same polarity is to be designed, an electronic VCMA larger by more than a factor of 10 would be feasible. In ordered alloys, it is reported that both the orbital magnetic moment and magnetic dipole moment can be modified by controlling a chemical order⁵⁵ Hence, it is expected that the VCMA mechanism is also controlled in the same manner. A material with both VCMA in the same polarity should be found in such variety kinds of ordered alloy.

- For applications, the biggest question will be how material specific this effect is. The L10 structure used in this paper is not really suitable for MRAM given the low TMR values which would be expected in a magnetic tunnel junction built from this material. For example if the authors used 3 monolayers of Fe | 1 monolayer of Pt, or if the structure was 001 instead of 111, would they see a similar effect?

- On a related note, how sensitive is this effect to Pt thickness? Can a dusting layer with nominal thickness less than one monolayer accomplish the same goal?

- In general, I think the latter two questions illustrate a need for control experiments with varying thicknesses and materials, to evaluate the robustness and practical application potential of this effect.

For the “3 monolayers of Fe | 1 monolayer of Pt”, we have already conducted a similar controlled experiment with (001)-oriented epitaxial multilayer, Fe(0.5 nm)|Pt(0.2)|MgO(2). Figure R1 shows the voltage-controlled magnetic anisotropy, where external voltages of ± 2.6 V (± 0.175 V nm⁻¹) were applied. The vertical axis shows the magnetisation curve measured by XMCD of the Pt-L₃ edge. A magnetic field was applied perpendicular to the film plane. The magnetic anisotropy energy change in the system was estimated to be 0.14 pJ V⁻¹ m⁻¹, whereas 0.03 pJ V⁻¹ m⁻¹ was obtained in our previous study with Fe(0.5 nm)|MgO(2)|SiO₂(5) 001-oriented epitaxial multilayer. [Ref. 35] The tendency of the Pt|MgO interface to exhibit a VCMA several times larger

than that of Fe|MgO was reproduced. For crystal orientation, we have already employed (001)-oriented $L1_0$ -FePt, which can be matched to conventional MgO-based magnetic tunnel junctions. However, as the reviewer mentioned, it is also important to obtain high TMR for MRAM. In this regard, we should think that the material design exhibits large quadrupole effects with $3d$ transition metals. We would like to conduct such material research in the future.

Fig. R1 (a) Sample structure for controlled experiments. (b) Magnetisation hysteresis curve under perpendicular magnetic field.

For the Pt thickness dependence, we also conducted a controlled experiment with (001)-oriented Fe|Pt(t)|MgO epitaxial multilayer. Although the experiment was rough ($t = 0.1, 0.2, 0.4, 0.8$ nm), the VCMA was maximized at $t = 0.2$ nm. Hence, less than one monolayer of coverage (< 0.2 nm) by the Pt reduces the VCMA. However, this is also the case with chemically ordered FePt. In the case of chemically disordered alloys, which might induce different orbital and dipole contributions, less than one monolayer of coverage may cause a VCMA maximum.

For the FePt thickness dependence, the results for the VCMA are summarized in Supplementary Information 1. However, it is difficult to conduct its XMCD experiment. When we increase the FePt layer thickness, both interfacial Pt, where an external electric field is applied, and Pt layers also exhibit its XMCD. Therefore, it is difficult to detect its voltage-induced XMCD signal within our limited resources of synchrotron experiment. Instead, we conducted a controlled experiment with the aforementioned epitaxial multilayer: Fe (0.50 nm)|Pt (0.20 nm)|MgO (2 nm). As discussed above, the VCMA in the control device exhibits a similar effect as compared with the FePt|MgO system. Moreover, a similarly induced XMCD was also observed. The induced XMCD is comparable to the $L1_0$ -FePt in Fig. 2b in the main text if the induced XMCD is scaled with an electric field and a number of Pt monatomic layers. This means that only the interfacial Pt has an important role, and the VCMA at the Pt|MgO interface should be

robust to the total thickness variations. This is consistent with the first-principles study shown in Fig. 4 in the main text.

For material dependence, although we expect that other *5d* transition metals show similar effects, we have not yet confirmed these. This is because the fabrication technology of such an alloy with a well-defined structure has not been established yet. This requires many resources and can be conducted as future work.

We would like to add the following modification to our manuscript. Thank you for the many valuable comments.

REVISION 11

(P13L326*, *after* revision)

To evaluate the robustness of the present study, a controlled experiment with (001)-oriented epitaxial multilayers of Fe(0.5 nm)|Pt|MgO was conducted. (See Supplementary Information 1). First, the VCMA was sensitive to the Pt layer thickness and was almost maximized when one Pt monatomic layer (0.2 nm) was employed. Second, an induced-XMCD similar to that shown in Fig. 2 was also observed, and it was scaled with an external electric field and a number of Pt monatomic layers. Hence, the one monatomic layer of Pt at the Fe|MgO interface contributes the most to the enhanced VCMA in the system. This is consistent with the first-principles study shown in Fig. 4. In the chemically ordered FePt alloy, one monatomic layer of coverage by the Pt is the best condition for VCMA; however, less than one monatomic layer of coverage may exhibit their VCMA maximum in the case of chemically disordered alloys, which should show different orbital and dipole contributions⁵⁵.

REVISION 12

(Supplementary Information: P03L47, *after* revision)

As the aforementioned FePt|MgO systems are perpendicularly magnetised at every thickness, it is difficult to characterize the VCMA energy. As shown in Supplementary Fig. 2a, we also conducted a controlled experiment with the following epitaxial multilayer: MgO(001) substrate|MgO buffer (5 nm)|V (30 nm)|Fe (0.50 nm)|Pt (0.20 nm)|MgO barrier (2 nm)|SiO₂ barrier (5 nm)|Cr (2 nm)|Pd (5 nm). Supplementary Figure 2b shows the VCMA of the multilayer measured by XMCD at the Pt-*L*₃ edge under a magnetic field perpendicular to the film plane. External voltages of ± 2.6 V correspond to electric fields of ± 0.175 V nm⁻¹ in the MgO dielectric. The multilayer shows a VCMA of 0.14 pJ V⁻¹ m⁻¹. Hence, we could expect similar VCMA energy in *L*₁₀-FePt|MgO systems. As compared with our previous study¹, the Pt|MgO interface

($0.14 \text{ pJ V}^{-1} \text{ m}^{-1}$) shows a VCMA several times larger than that of Fe|MgO ($0.03 \text{ pJ V}^{-1} \text{ m}^{-1}$).

- Can the authors rule out electrochemical effects, given the slow measurement method and the very large voltages applied?

Although the external voltage was $\pm 200 \text{ V}$, the external electric field at the Pt|MgO interface was about $\pm 0.7 \text{ V nm}^{-1}$. As $\pm 1 \text{ V nm}^{-1}$ can be feasible in MgO-based magnetic tunnel junction devices, the external electric field of $\pm 0.7 \text{ V nm}^{-1}$ was not very large. Moreover, because the induced effective spin magnetic moment of Pt ($> 10\%$) cannot be scaled by the induced hole in d -orbitals (1%), the obtained results was not caused by the substantial change in the $5d$ valence state of the Pt, which indicates an electrochemical effect. We would like to include additional discussions in the main text as follows.

REVISION 13

(P07L178, *before* revision)

Note that the change in the effective spin magnetic moment (13%) was an order of magnitude larger than the change in the hole number (1%). Therefore, the change was not explained by voltage-induced redox reaction but by the electron redistribution in the Pt atom.

(P07L178, *after* revision)

Note that the change in the effective spin magnetic moment (13%) was an order of magnitude larger than the change in the hole number (1%). Therefore, the change cannot be explained by the voltage-induced electrochemical reaction resulting in substantial changes in a valence state. In addition, the change should be attributed to the electron redistribution in the Pt atom.

- It would be helpful to applied researchers and device physicists if the units of VCMA were put in the usually used ones, such as anisotropy energy change per electric field. Also, it may help to move some of the information from Supplementary section I to the main manuscript text.

Thank you for the suggestion. As in the aforementioned discussion, the VCMA at the Pt|MgO interface is $0.14 \text{ pJ V}^{-1} \text{ m}^{-1}$. We would like to include additional discussions

both in the main text and Supplementary Information.

REVISION 14

(P04L89, *before* revision)

An $L1_0$ -FePt|MgO system has been prepared as shown in Fig. 1b (See Methods 1.). The FePt ferromagnetic layer consists of two monatomic layers of Fe and Pt. Figure 1c shows a high-angle annular dark-field scanning transmission electron microscopy (HAADF-STEM) image of the sample (See Methods 2.). This image shows that the FePt is epitaxially grown on Pd(001) and that a monoatomic Pt layer lies between the Fe and MgO. We studied voltage-induced changes in the electronic and magnetic states of Pt, which has a large spin-orbit interaction and should be responsible for the large VCMA in the system (See Supplementary Information 1 and Ref. 17). Using the experimental setup shown in Fig. 1b, we applied external voltages of ± 200 V to the FePt|MgO tunnel junction. X-ray absorption spectroscopy (XAS)/XMCD spectra were recorded in situ at the Pt L_3 and L_2 edges by detecting the X-ray fluorescence yields (See Methods 3.). In this configuration, positive external voltages induce electron accumulations at the Pt|MgO interface.

(P04L89, *after* revision)

An $L1_0$ -FePt|MgO system has been prepared to conduct XMCD spectroscopy. First, a controlled experiment was conducted to optimize the structure of $L1_0$ -FePt|MgO (See Method 1 and Supplementary Information 1). As suggested by the previously reported theoretical study¹⁸, the VCMA in the FePt|MgO with Pt|MgO interface ($0.14 \text{ pJ V}^{-1} \text{ m}^{-1}$) was several times larger than that with the Fe|MgO interface ($0.03 \text{ pJ V}^{-1} \text{ m}^{-1}$). Moreover, more than one monatomic layer of Fe is necessary to induce a ferromagnetic property in the FePt. Hence, the FePt ferromagnetic layer consisting of two monatomic layers of Fe and Pt, which has a Pt|MgO interface, was designed as shown in Fig. 1b. Figure 1c shows a high-angle annular dark-field scanning transmission electron microscopy (HAADF-STEM) image of the sample (See Method 2). This image indicates that the FePt is epitaxially grown on Pd(001) and a monoatomic Pt layer lies between the Fe and MgO. We studied voltage-induced changes in the electronic and magnetic states of Pt, which exhibits a large spin-orbit interaction and should be responsible for the large VCMA in the system. Using the experimental setup shown in Fig. 1b, we applied external voltages of ± 200 V to the FePt|MgO tunnel junction. The external voltage is equivalent to an external electric field of ± 0.7 V/nm in the MgO dielectric. Such an electric field magnitude is often used in MgO-based magnetic tunnel junctions²³. X-ray absorption spectroscopy (XAS) and its XMCD

spectra were recorded in situ at the Pt L_3 and L_2 edges by detecting the X-ray fluorescence yields (See Method 3). In this configuration, positive external voltages induce electron accumulations at the Pt|MgO interface.

REVISION 15

(Figure 1 caption, *before* revision)

(d) Element-specific magnetisation hysteresis curves from the XMCD signal at the L_3 edge energy of Pt under external voltages of ± 200 V.

(Figure 1 caption, *after* revision)

(d) Element-specific magnetisation hysteresis curves from the XMCD signal at the L_3 edge energy of Pt under external voltages of ± 200 V. The external voltage is equivalent to an external electric field of ± 0.7 V nm⁻¹ in the MgO dielectric. Such an electric field magnitude is often used in MgO-based magnetic tunnel junctions.

see **REVISION 12**.

- Occasional typos and grammatical errors should be fixed.

We have carefully checked and modified our manuscript. Thank you for the comment.

Response to Reviewer #3

We would like to thank Reviewer #3 for his/her precise review as well as a high evaluation of our manuscript. The number of valuable comments were very useful for the revision. We have revised the manuscript as follows.

Voltage controlled magnetic anisotropy (VCMA) holds the key to the next generation ultra-low energy spintronics devices. The authors discovered a new mechanism for VCMA in FePt/MgO structure. In addition to the conventional contribution to VCMA due to voltage induced modification of orbital moments, the author identified a new source that is related to the magnetic dipole moment produced by the electric quadrupole. The change of magnetic anisotropy energy (MAE) can be larger with the dipole effect, which could lead to a potential large enhance of the VCMA effect. This work is novel, important and will be interesting a wide range of readers. Therefore this study is suitable for a premium journal like Nature Communications, if the authors can address my following questions:

1. This work highlights the importance of Pt in obtaining the large contribution to VCMA as described by the second term of Eq. 5. However it is know if Pt is directly placed next to MgO, the tunneling magnetoresistance (TMR) of the junction will be diminished. Would this quadrupole induced effect also exist in normal FM such as Fe, Co and Ni? Because if it only exists with heavy metals with large spin orbit interaction, it will inevitably destroy the TMR therefore making this VCMA mechanism less useful.

Thus far, the quadrupole-induced effect has not been considered in normal $3d$ transition metals. This is because spin-orbit coupling in such metals is small, and the contribution of the second term in Fig. 5 to the perpendicular MAE is negligible. However, the suggestion by the reviewer is very attractive. We should think that the material designs exhibit large quadrupole effects with $3d$ transition metals. We would like to conduct such important studies in the near future.

2. It is discovered the VCMA effects due to orbital moment and induced magnetic dipole moment cancel each other. It is speculated that the VCMA efficiency can be dramatically improved if the signs of the two effects are the same. Can the authors discuss under what situations we might expect the signs to be the same?

Thank you for the valuable comment. As discussed in ref. [P. Kamp *et al.*, Phys. Rev. B 59, 1105 (1999)], the orbital magnetic moment and magnetic dipole moment can be modulated by controlling a chemical order. Hence, it is expected that the VCMA mechanism can be also controlled in the same manner. A material with both VCMA mechanisms adding up should be found in a variety of ordered alloys.

see **REVISION 10**

3. What is the VCMA effect in the present study in terms of pJ/Vm?

We conducted a controlled experiment with Fe(0.5 nm)|Pt(0.2)|MgO(2) 001-oriented epitaxial multilayer to quantitatively characterize the VCMA at the Pt|MgO interface. We obtained a VCMA effect of $0.14 \text{ pJ V}^{-1} \text{ m}^{-1}$. We would like to modify the main text as follows.

see **REVISION 12**

see **REVISION 14**

4. What is the uncertainty of Δn_{total} in the table 1?

Thank you for the suggestion. We employed 9.8 for the dielectric constant of MgO; this is the literature value of MgO bulk. It is reported that the dielectric constant of MgO thin film depends on the film quality and may be smaller than that of bulk. Hence, δn_{total} can be overestimated. We would like to modify our manuscript as follows.

REVISION 16.

(Supplementary Information: P10L154, *before* revision)

Then, the result of subtracting the δn_{total} of +200 V from the δn_{total} of -200 V in the FePt|MgO|polymide was estimated to be +0.06.

(Supplementary Information: P10L154, *after* revision)

Thus, the result of subtracting the δn_{total} of +200 V from the δn_{total} of -200 V in the FePt|MgO|polymide was estimated to be +0.06. As mentioned above, we employed the dielectric constants of bulk for the calculation. As the dielectric constant of thin film depends on the film quality and is likely smaller than that of bulk, the δn_{total} might be overestimated. The error can be, at most, 10–20%.

5. The format of citations (line 57-63) is not consistent.

Thank you for the suggestion. We have modified the format.

Reviewers' Comments:

Reviewer #1 (Remarks to the Author):

The authors have addressed satisfactorily all my comments and suggestions and have revised the manuscript accordingly. Consequently, the manuscript does warrant publication in Nature Communications.

Reviewer #2 (Remarks to the Author):

The authors have provided satisfactory answers to all my questions from the previous review. I would like to recommend this paper for publication.

Reviewer #3 (Remarks to the Author):

The authors have sufficiently addressed my comments, and the questions raised by other referees. This is an important work and now it has my recommendation for publication on Nature Communications.